# Adherence to a Supplemented Mediterranean Diet Drives Changes in the Gut Microbiota of HIV-1-Infected Individuals

**DOI:** 10.3390/nu13041141

**Published:** 2021-03-30

**Authors:** Roque Pastor-Ibáñez, Juan Blanco-Heredia, Florencia Etcheverry, Sonsoles Sánchez-Palomino, Francisco Díez-Fuertes, Rosa Casas, María Ángeles Navarrete-Muñoz, Sara Castro-Barquero, Constanza Lucero, Irene Fernández, Lorna Leal, José Miguel Benito, Marc Noguera-Julian, Roger Paredes, Norma Rallón, Ramón Estruch, David Torrents, Felipe García

**Affiliations:** 1AIDS Research Group, IDIBAPS, Hospital Clinic, University of Barcelona, 08036 Barcelona, Spain; rpastor@clinic.cat (R.P.-I.); MFETCHEV@clinic.cat (F.E.); ssanchez@clinic.cat (S.S.-P.); frandifu@icloud.com (F.D.-F.); MCLUCERO@clinic.cat (C.L.); IFERNANDEZC@clinic.cat (I.F.); LALEAL@clinic.cat (L.L.); 2IrsiCaixa AIDS Research Institute, Germans Trias i Pujol University Hospital, 08916 Badalona, Spain; jblancoh@irsicaixa.es; 3Germans Trias i Pujol Research Institute (IGTP), 08916 Badalona, Spain; 4Barcelona Supercomputing Center, 08034 Barcelona, Spain; 5Infectious Diseases Department, Hospital Clínic, IDIBAPS, University of Barcelona, 08036 Barcelona, Spain; 6Department of Internal Medicine, Hospital Clinic, Institut d’Investigació Biomèdica August Pi i Sunyer (IDIBAPS), University of Barcelona, Villarroel, 170, 08036 Barcelona, Spain; RCASAS1@clinic.cat (R.C.); SACASTRO@clinic.cat (S.C.-B.); RESTRUCH@clinic.cat (R.E.); 7CIBER 06/03: Fisiopatología de la Obesidady la Nutrición, Instituto de Salud Carlos III, 28029 Madrid, Spain; 8HIV and Viral Hepatitis Research Laboratory, Instituto de Investigación Sanitaria-Fundación Jiménez Díaz, Universidad Autónoma de Madrid (IIS-FJD, UAM), 28040 Madrid, Spain; mangelesnm94@gmail.com (M.Á.N.-M.); jbenito1@hotmail.com (J.M.B.); normaibon@yahoo.com (N.R.); 9Hospital Universitario Rey Juan Carlos, Móstoles, 28933 Madrid, Spain; 10Hospital Universitari Germans Trias i Pujol, IrsiCaixa, 08916 Badalona, Spain; mnoguera@irsicaixa.es (M.N.-J.); rparedes@irsicaixa.es (R.P.); 11Computational Genomics Groups, Barcelona Supercomputing Center (BSC), 08034 Barcelona, Spain; david.torrents@bsc.es; 12Institució Catalana de Recerca i Estudis Avançats (ICREA), 08010 Barcelona, Spain

**Keywords:** gut microbiota, supplemented Mediterranean diet, lipids, lymphocyte subsets, HIV-1, Treg cells

## Abstract

Objective: The health effects of a supplemented Mediterranean diet (SMD) with extra-virgin olive oil (EVOO) and nuts are well documented in non-HIV-infected individuals. We hypothesised that the benefits of an SMD could be mediated by changes in the gut microbiota, even in those with an altered intestinal microbiota such as people living with HIV. Design: Individuals living with HIV (*n* = 102) were randomised to receive an SMD with 50 g/day of EVOO and 30 g/day of walnuts (SMD group) or continue with their regular diet (control group) for 12 weeks. Methods: Adherence to the Mediterranean diet was assessed using the validated 14-item MD-Adherence-Screener (MEDAS) from the PREDIMED study. A sub-study classifying the participants according to their MEDAS scores was performed. Results: The lipid profile was improved in the SMD group vs. that in the control group (delta-total cholesterol and delta-B-lipoprotein). The immune activation (CD4+HLADR+CD38+ and CD8+HLADR+CD38+ cells) and IFN-γ-producing T-cells significantly decreased at week 12 compared to the baseline in the SMD group but not in the control group. The gut microbiota in those from the high-adherence group presented significantly high diversity and richness at the end of the intervention. *Succinivibrio* and *Bifidobacterium* abundances were influenced by the adherence to the MD and significantly correlated with Treg cells. Conclusion: The Mediterranean diet improved metabolic parameters, immune activation, Treg function, and the gut microbiota composition in HIV-1-infected individuals. Further, Mediterranean diet increased the *Bifidobacterium* abundances after the intervention, and it was associated to a beneficial profile.

## 1. Background

Antiretroviral therapy (ART) has contributed to a significant decrease in the morbidity and mortality associated with acquired immune deficiency syndrome (AIDS) in human immunodeficiency virus 1(HIV-1) infected individuals [1]. However, people with chronic HIV infection have an increased risk of death due to non-AIDS events when compared to non-infected individuals [2,3,4]. This increase in morbidity and mortality is related to cardiovascular diseases, malignancies, dementia, osteoporosis, and also a rise in comorbidities such as high blood pressure, diabetes, bone fractures, and kidney failure [5]. The mechanisms driving these non-AIDS events are chronic immune activation and inflammation. Immune activation is a good independent predictor of morbidity and mortality in HIV infection [6]. It has been suggested that microbial translocation from the intestinal lumen to the systemic circulation plays a key role in the persistence of immune activation [7,8]. The destruction of gut-associated lymphoid tissue (GALT) has been postulated to underlie this microbial translocation. A rapid depletion of CD4+ T-cells at mucosal sites takes place during the first weeks of infection and persists in chronic infections. Regarding the composition of the microbiota, some studies suggest that an HIV-1 infection induces dysbiosis in the gut composition, reducing microbial richness and diversity [9,10]. It is evident that the composition of the microbiota depends on nutritional habits. Bacteria that ferment proteins produce some toxic products that drive inflammatory processes, while microbes that ferment fibre inhibit inflammation [11]. The Mediterranean diet (MD) is commonly characterised by a high consumption of olive oil (OO), fruits, vegetables, legumes, nuts, and whole grain cereals, as well as a moderate consumption of dairy products, fish, poultry meat, and red wine, accompanied by a low intake of red meat, processed meat, and sugars [12]. Extra virgin olive oil (EVOO) and walnuts contain bioactive compounds, such as monounsaturated fatty acids (MUFAs); α-tocopherol; polyunsaturated fatty acids (PUFAs), e.g., omega-3 and omega-6 fatty acids; polyphenols; and other phytochemicals, which have antioxidant and anti-inflammatory properties [13]. Data published to date and from the PREDIMED study have revealed that an MD supplemented with EVOO or nuts delays the appearance of cardiovascular disease, atrial fibrillation, cancer, and neurodegenerative disease in high-risk populations [14,15]. Moreover, the MD induces shifts in the gut microbiota [16]. 

However, there is a lack of robust evidence linking the effects of an MD on microbiota composition in HIV-1-infected individuals receiving ART. This information could help us understand the relationship between HIV infection and the immune-activation-driven non-AIDS events. Thus, the main aim of this study was to evaluate the composition of the microbiota of HIV-1-infected subjects based on their adherence to the supplemented Mediterranean diet (SMD), correlating this with clinical parameters, bacterial translocation, inflammatory markers, and different T-cell subsets.

## 2. Methods

### 2.1. Study Design

A single-centre randomised controlled open-label clinical study was performed. HIV-infected individuals on successful ART were randomised to either receive the SMD or continue with their regular diet (control group) for 12 weeks. The inclusion criteria are shown in the Appendix A. 

At the time of enrolment and after 12 weeks at follow-up, data on the demographic and clinical variables were collected for each participant (see Appendix A).

After the participants provided their signed informed consent, they were randomly assigned to one of two study groups: the SMD group receiving the MD supplemented with EVOO and walnuts or the control group continuing with their regular diet. Randomisation was performed by random assignment. Participants in the SMD group were given supplementary foods at no cost: EVOO (1 L/week) and walnuts (30 g/day). 

The validated 14-item MD Adherence Screener (MEDAS) from the PREDIMED study was used, which produces a score ranging from 0 to 14, with a higher score indicating better adherence [14]. All participants had a face-to-face interview with a dietitian at baseline and after 12 weeks to determine their adherence to the intervention. The MEDAS score was obtained at baseline (week 0) and at the end of the intervention (week 12). However, only the participants in the SMD group received a thorough dietary and nutritional assessment. All the study participants were classified into three groups based on their adherence to the MD as follows: those with a MEDAS ≥ 10 points were placed in the high-adherence group, those with a MEDAS ≥ 7 points and MEDAS < 10 points were placed in the medium-adherence group, and those with a MEDAS < 7 points were put in the low-adherence group. 

All participants provided written informed consent before entering the study. The study was approved by the Institutional Review Board of the Hospital Clinic of Barcelona (Spain). The trial was registered (NCT03846206).

### 2.2. Nutritional Biomarkers and Polyphenols

To ensure diet compliance, both at baseline and after 12 weeks, α-linolenic acid levels (as a measure of adherence to walnut consumption recommendations) in plasma as well as tyrosol and hydroxytyrosol levels (as a measure of adherence to EVOO consumption recommendations) in urine were assessed by gas chromatography coupled with mass spectrometry. In addition, dietary polyphenol intake was estimated as described elsewhere [17]. 

### 2.3. Markers of Inflammation and Bacterial Translocation

Ultrasensitive C-reactive protein (uCRP) levels were determined with a latex enhanced immunoturbidimetric assay (Advia 2400; Siemens Diagnostics, Germany). Interleukin-6 (IL-6) levels were measured by an immunoenzymatic assay with solid-phase amplification on microtitration plates (DIAsource, Nivelles, Belgium). D-dimer levels were measured with a turbidimetric method (Innovance; Siemens Diagnostics, Marburg, Germany) in a BCS XP (Siemens Diagnostics) automated coagulation system. Bacterial translocation was evaluated by measuring the levels of lipopolysaccharide-binding protein (LBP), sCD14, and EndoCAb in peripheral blood using the enzyme-linked immunosorbent assay (ELISA) technique.

### 2.4. T-Lymphocyte Subsets

Activation status and Th17 and Treg cell subsets were analysed by multiparametric flow cytometry using the Cytomics FC500 flow cytometer (Beckman Coulter). For this purpose, three different panels of monoclonal antibodies were designed to stain peripheral blood mononuclear cells (PBMCs). Each panel included five different monoclonal antibodies to measure the level of activation of CD4+ and CD8+ T lymphocytes, as well as Th17 cell subset levels and Treg cell subset levels. Th17 cells were defined as CD4+ T-cells producing IL17A, while Treg cells were defined as CD4+ T-cells that were FoxP3+CD25+CD127−. Data analysis was performed using the CXP software (Beckman Coulter). The complete methodology is detailed in the Appendix A.

### 2.5. Microbiota Composition

Faecal DNA extraction: Faecal samples from the study participants were collected using the SOP.03.V2 protocol from the International Human Microbiome Standards (IHMS). The handbook included in the QIAamp DNA Stool Mini Kit was used with some modifications (see the Appendix A). Samples were stored at −20 °C until analysis.

### 2.6. Bioinformatics and Statistical Analysis

In order to study the effect of the intervention, we used two different approaches. First, we focused on the analysis comparing the different time points (basal vs. week 12) from each randomisation group (SMD and controls). Next, the analysis was focused on the mathematical change using the delta (Δ) of each parameter by randomisation group (ΔSMD vs. Δcontrols). Δ was calculated from each marker as follows: Δ = (value at week 12) − (value at baseline).

In order to study the effect of the diet supplementation, we used the ΔMEDAS between baseline and week 12, (end of the study). Both Δ of the quantified markers (metabolic, microbial translocation, inflammatory, nutritional, and immunological markers) and Δ of the gut microbiota composition were analysed. 

Due to the different number of patients included in the high- and low-adherence groups at baseline and at the end of the study, and with the aim of avoid bias induced by the inclusion of the same subject in two different groups, we established the adherence groups as fixed by the MEDAS score at the end of the study (week 12). In order to study the effect of the nutritional behaviour in the gut microbiota composition, we used the MEDAS score at week 12, using both time points (baseline and week 12) from each participant; in this regard, the two time-point samples from each patient were considered as replicates, and each replicate was considered as a dependent variable in the statistical analysis to avoid bias due to subject effect.

The statistical analysis for the different parameters (lipid, metabolic, nutritional profile, inflammation, bacterial translocation, immune system activation, Treg, and Th17 T-lymphocyte subsets) between the different groups and time points was performed using the R software. Differences between groups were assessed with multiple Mann–Whitney–Wilcoxon tests or the Kruskal–Wallis test with Bonferroni correction, as required. 

The gut microbiota composition was assessed using the QIIME2 software [18]. The 16S rRNA sequence analysis is detailed in the Appendix A. Microbiome samples were clustered according to their genera composition using a non-metric multidimensional scaling (NMDS) approach based on ecological distance matrices calculated by Bray–Curtis dissimilarities, implemented in R packages (Vegan, metaMDS, and ggplot2 packages). NMDS ellipses were drawn based on a confidence interval (CI) of 0.95. 

For the determination of the alpha diversity, we used the following diversity indices (Shannon index, 1/Simpson, and Fisher index) and richness parameters (observed). For the determination of the beta diversity, we used the principal coordinate analysis (PCoA) with the NMDS obtained, as well as the gut microbiota composition (relative abundances). The differences in gut microbiota composition between groups were inspected using the linear discriminant analysis (LDA) effect size (LEfSe). Sample dissimilarity between groups was evaluated using the Adonis test (PERMANOVA). 

The association between variables (metabolic, microbial translocation, inflammatory, nutritional, and immunological markers) and the different groups (randomisation groups: control or SMD; adherence groups: low or high adherence; time points: baseline or week 12) was evaluated using the two-tailed Wilcoxon rank sum test, when two groups were compared (paired: basal vs. week 12; unpaired: low-adherence vs. high-adherence). When three or more groups were compared, it was assessed with the Kruskal–Wallis test with Bonferroni correction. Correlations between the markers and the gut microbiota composition were obtained using the Spearman’s rank correlation coefficient with Holm’s correction for multiple comparisons. Data were considered statistically significant at *p* ≤ 0.05.

In the post hoc analysis, individuals in the group men who have sex with men (MSM) (*n* = 60) were selected to evaluate the similarities between bacterial communities. Balances between subgroups were obtained using the gneiss plugin from QIIME2, next steps were executed in R software. The cross-validation in balances was conducted in the selbal analysis as described by Rivera-Pinto et al. [19].

## 3. Results

### 3.1. Demographic Data

In this study, 102 participants living with HIV were randomised. Twenty participants only attended the visit at baseline and were thus excluded from the analysis. Therefore, data from 82 participants who completed the assessment in week 12 are shown (Appendix A). No differences were detected at baseline between patients who were lost to follow-up and those who completed the evaluation in week 12 (data not shown). The median age of the participants was 47 years, 69 (84%) were men, and 60 (73%) were men who have sex with men (MSM). The median (interquartile range (IQR)) CD4+ T-cell count at baseline and at the nadir was 820 (663, 1028) and 378 (312, 468) cells/µL, respectively. Appendix A shows the characteristics of the subjects by study group.

### 3.2. Adherence to the SMD Stratified by Randomisation Group 

As shown in Table 1, the distribution of patients in the different adherence groups was similar at baseline. Individuals in the SMD group (*n* = 37) significantly improved their nutritional habits from baseline to week 12 (from a median MEDAS score of 6 to 12 points, *p* < 0.005; see Appendix A). At baseline, only 8% (*n* = 3) of the individuals in the SMD group (*n* = 37) presented a high adherence to the MD (MEDAS ≥ 10 points), with this proportion increasing to 84% (*n* = 31) at the end of the study (week 12). Conversely, individuals in the control group (*n* = 37) maintained their baseline MEDAS scores at the end of the study. The SMD group significantly increased their consumption of EVOO, nuts, walnuts, fruits, vegetables, legumes, and fish (*p* < 0.05). In addition, they also reduced their intake of refined cereals, red meat, and meat products, as well as pastries, cakes, and sweets (*p* < 0.05). No differences were found in the key food items analysed in the individuals of the control group (data not shown).

### 3.3. Metabolic, Microbial Translocation, Inflammatory, and Immunological Parameters 

An improvement in the lipid profile was observed in the SMD group compared to that in the control group ((mean (standard error, SE) delta total cholesterol −3.36 (3.29) vs. +9.43 (4.51) mg/dL, respectively, *p* = 0.02) (mean (SE) delta B-lipoprotein −2.11 (3.34) vs. +7.75 (3.58) mg/dL, respectively, *p* = 0.05)) A trend was observed in the increase of the CD4+ T-cells in the SMD at the end of the study (CD4+ (%), *p* = 0.07). IFN-γ-producing T-cells significantly decreased at week 12 with respect to baseline in the SMD group (median (IQR) CD8+IFNγ+: 61.51 (49.51, 69.35) and 50.49 (40.93, 68.1) for week 0 and 12, respectively, *p* = 0.002). No significant differences were observed between the SMD and control groups in the markers of inflammation, microbial translocation, and immune activation examined (Appendix A). 

### 3.4. Gut Microbiota Diversity and Richness

No differences in alpha and beta diversity were found between the study groups at the end of the study. By contrast, we found important differences in alpha diversity (observed, *p* = 0.002; Fisher, *p* < 0.005) and beta diversity between the men and women participants (Figure 1; Adonis test (PERMANOVA): *p* = 0.001, R^2^ = 0.18) at baseline, indicating that the gut microbiota of the studied cohort was highly influenced by the sex. 

## 4. Sub-Study with MSM Individuals Stratified by the MEDAS Score

Although we did not find differences in the gut microbiota between the study groups, we hypothesised that patients with a higher adherence to the MD could have a better profile. To assess this hypothesis, we performed a sub-study, classifying the participants according to their MEDAS score (MEDAS ≥ 10 points indicated high adherence, 7 points ≤ MEDAS < 10 points indicated medium adherence, and MEDAS < 7 points indicated low adherence). The groups were fixed using the MEDAS score obtained at the end of the study (week 12), this was done in order to avoid bias induced by the inclusion of the same subject in two different groups. In order to study the effect of the diet supplementation we used the ΔMEDAS between baseline and week 12, (end of the study). We classified the subjects by the increase in their MEDAS scores from baseline to the end of the study (ΔMEDAS < 2 points or ΔMEDAS ≥ 4 points). We excluded the women participants and focused on MSM individuals (*n* = 60), given the significant differences observed in the gut microbiota between men and women and also the previous results about the impact of sexual preference on the microbiome [10]. As shown in Table 1, MSM individuals presented the same adherence behaviour according to the randomisation arm as the whole cohort. The key food and dietary nutrient intake among the MSM individuals stratified by their MEDAS scores at week 12 are shown in Appendix A. 

### 4.1. Metabolic, Microbial Translocation, Inflammatory, and Immunological Parameters Stratified by the MEDAS Scores 

We divided participants at baseline according to Table 1, to see whether the adherence to MD at the basal time point also affected the inflammatory, bacterial translocation, and immunological markers. The data showed no statistically significant differences between the low-adherence and the high-adherence groups (Appendix A). The analysis of the entire cohort by the adherence groups (*n* = 74) indicated that B-lipoprotein levels increased significantly between week 0 (baseline) and week 12 (end of the study) in the low-adherence group (mean (SE) delta B-lipoprotein +9.70 mg/dL (3.55), *p* = 0.02). Additionally, there was a significant reduction of immune activation in the high-adherence group (mean (SE) delta CD4+HLADR+CD38+ −0.16% (0.67), *p* = 0.02; delta CD8+HLADR+CD38+ −0.26% (1.46), *p* = 0.01), in contrast to the low-adherence group, which showed an increase (mean (SE) delta CD4+HLADR+CD38+ +1.00% (1.91); delta CD8+HLADR+CD38+ +0.38% (2.41)).

### 4.2. Metabolic, Microbial Translocation, Inflammatory, and Immunological Parameters in MSM Stratified by Their MEDAS Scores

No important differences were observed in the metabolic, inflammatory, and translocation markers in the MSM individuals. Regarding immune activation, individuals in the high-adherence group presented a significant reduction in their CD4+HLADR+CD38+ (*p* = 0.003) and CD8+HLADR+CD38+ (*p* = 0.007) cells compared to the low-adherence group (see Appendix A). In this case, there were no significant changes in the CD8+IFNγ+ cells. Even so, the CD8+IFNγ+ cells presented a direct correlation with CD4+HLADR+CD38+ (*p* = 0.01, R = 0.46) and CD8+HLADR+CD38+ (*p* = 0.003, R = 0.23) cells in the MSM subjects in the high-adherence group (not shown).

### 4.3. Gut Microbial Diversity and Richness

After stratifying the MSM individuals by their MEDAS scores from week 12, the gut microbiota from individuals in the high-adherence group presented significantly higher species counts and richness (Figure 2a). Beta diversity was examined using the principal coordinates analysis (PCoA, Figure 2b). All obtained PCoAs both for the entire cohort and the MSM individuals at baseline and week 12 are shown in Appendix A. To inspect beta diversity, different statistical approaches were undertaken using the obtained distances. The PERMANOVA analysis showed important differences between the groups: high-adherence group vs. low-adherence group (*p* < 0.005, Q < 0.05) and the medium-adherence group vs. the low-adherence group (*p* < 0.05, Q < 0.05). These results were confirmed by the Adonis test (*p* < 0.05, R = 0.022) and the multivariate linear regression modelling (*p* = 0.05; adjusted R-squared: 0.03; t-statistic: medium adherence < 0.05, high adherence < 0.05). 

### 4.4. Gut Microbiota Composition

No differences were found in the gut microbiota composition between the study groups (the SMD and control groups, Appendix A). The composition of the gut microbiota was assessed in the subgroup of MSM individuals living with HIV stratified by their MEDAS scores from week 12. The abundance of *Bacteroides* was lower in the high-adherence group than in the low-adherence group (*p* = 0.0001, Appendix A). 

The linear discriminant analysis (LDA) effect size (LEfSe) of the basal differences between low- and high-adherence groups are shown in Figure 3a. When both time points were considered, LEfSe analysis showed that the individuals in the high-adherence group presented significantly increased levels of: *Burkholderiales*, *Butyrivibrio*, *Catenibacterium*, and *Succinivibrio*. Conversely, individuals in the low-adherence group presented high levels of: *Bacteroides, Parabacteroides, Desulfovibrio, Paraprevotella,* and *Bilophila* (Figure 3b). When the participants were studied according to the improvement in their MEDAS scores (ΔMEDAS) using the LEfSe analysis, the gut microbiota of the individuals with ΔMEDAS ≥ 4 points were enriched exclusively with the *Bifidobacterium* genus at week 12 compared to those who did not improve their MEDAS scores at week 12 (Figure 3c). 

Finally, the cross-validation in balances selection conducted in the selbal analysis (Figure 4) showed that *Bacteroides* (70%) and *Succinivibrio* (56%) were the bacterial genera that best explained the differences between the high-adherence and low-adherence groups (receiver operating characteristic (ROC) curve, area under the curve (AUC) = 0.842). 

### 4.5. Association between Changes in the Gut Microbiota Composition and Changes in the Metabolic, Microbial Translocation, Inflammatory, and Immunological Parameters

The potential associations between the microbiota composition and the parameters analysed in this study were explored in the MSM participants from the high-adherence group at week 12 (Figure 5). The prevalence of some of the bacterial genera (*Bacteroides*, *Blautia*, *Succinivibrio*, and *Butyrivibrio*) were influenced by the composition of the diet, particularly for those from the Firmicutes phylum. Saturated fat intake increased the levels of *Blautia*. In addition, an increase in the calorie (kcal) and carbohydrate intake was associated with a decrease in the levels of *Paraprevotella* and *Butyrivibrio* (data not shown). The relative abundance of the *Blautia* genus was positively associated with the levels of microbial translocation. The prevalence of the *Butyrivibrio* genus was inversely associated with immune activation and Treg cell subset levels, in contrast to some Proteobacteria (the *Succinivibrio* and *Desulfovibrio* genera) that directly correlated with immune activation and Treg cell subset levels. A similar analysis was carried out in the subgroup of MSM patients presenting a ΔMEDAS ≥ 4 points (Appendix A), which revealed that the abundance of the *Bifidobacterium* genus was inversely associated with the intake of saturated fats (data not shown). Finally, the relative abundance of the *Succinivibrio* genus was inversely associated with immune activation, while the abundance of the *Butyrivibrio* genus was inversely associated with CD4+IL17A+ T-cell levels.

## 5. Discussion

Dietary habits are one of the most important factors affecting the gut bacterial composition, richness, and diversity in the general population [20]. In individuals with a chronic HIV-1 infection, high-quality nutrition and food choices are important [21]. This study reflects the importance of nutritional advice that promotes a healthy and balanced diet. In the current study, 90% of the participants who had a dietary and nutritional assessment with an expert nutritionist improved their MEDAS scores at the end of the intervention. They significantly increased their consumption of vegetables, legumes, nuts, fruits, and fish, while reducing their intake of high-fat, processed, and unhealthy foods such as sweets and sugary beverages. There is a vast amount of scientific evidence showing that diets rich in unhealthy fats, sugars, and salt are closely associated with non-chronic diseases, such as cardiovascular diseases, diabetes, obesity, and cancer, as well as an increased mortality risk [22]. By contrast, healthy eating patterns such as the MD or similar diets with a high fibre content reduce these risks, as well as the risk of developing neurodegenerative diseases or atrial fibrillation [14,15]. The validated 14-item MEDAS questionnaire from the PREDIMED study reflected with high accuracy the nutritional behaviour of the individuals in this study and could be useful for clinicians aiming to identify nutritional risk [23]. It allowed us to detect the impact of the MD on the gut microbiota.

In this study, individuals who supplemented their diet with EVOO and walnuts improved their lipid profile, reducing their total cholesterol and B-lipoprotein levels. Healthy benefits associated with the MD may be explained by its high content of polyphenols. Polyphenols play an important role in health promotion, as they seem to show anti-inflammatory, antioxidant, antidiabetic, or anticarcinogenic effects, as well as improve lipid profile or adiposity. In the small intestine, the absorption of polyphenols is low (<10%), and the rest of the polyphenols can positively influence the microbiota composition. Nevertheless, the underlaying mechanisms through which polyphenols can exert these positive effects are as yet unknown [17,24,25]. It must be highlighted, that in our study, individuals with a high adherence to the MD, with high EVOO and walnut intake, improved their lipid profile, reducing their total cholesterol and B-lipoprotein levels. EVOO contains polyphenols, which possess vasodilatory, anti-thrombotic, anti-inflammatory and anti-apoptotic effects, leading to a cardioprotective state, as well as having antilipemic and anti-atherogenic effects [26,27].

Similar studies on individuals with HIV have described reductions in LDL cholesterol levels and systolic blood pressure after at least 24–48 weeks of being on the MD; however, no important changes in the inflammatory markers have been described [28,29]. It is remarkable that the inflammatory and bacterial translocation markers were not altered by the SMD. This could be due to the stable immune state of the recruited subjects and the limited duration of the dietary intervention. Consequently, longer dietary intervention could be useful to track changes in blood circulating inflammatory and bacterial translocation markers. Interestingly, we found lower levels of immune activation and IFN-γ-producing T-cells in the SMD group, suggesting a lower level of antigen-driven immune activation. In support of our findings, it is well described that a decrease in immune activation is critical for maintaining the integrity of mucosal barriers and the direct link between chronic immune activation and Th17 production [30,31,32]. Our results confirmed that the SMD was helpful in reducing the values of the metabolic and immune activation parameters associated with non-AIDS events in HIV-1-infected patients.

Briefly, we report a significant reduction in the levels of *Bacteroides* in individuals with a high adherence to the MD, affecting the *Prevotella/Bacteroides* ratio. In support of our findings, the *Bacteroides* enterotype has been described to be abundant in individuals with high protein and animal fat intakes [33], while the *Prevotella* genus is highly abundant in people with fibre-rich diets [34]. We also found that adherence to the MD influenced the composition of the microbiota. The participants with high adherence showed increases in the prevalence of *Burkholderiales*, *Butyrivibrio*, *Catenibacterium*, and *Succinivibrio*. Some of these genera have been observed to be increased in those with diets containing high amounts of fibre [35,36]. Finally, *Bifidobacterium* spp. have been reported to occur in the guts of individuals with diets rich in fibre, plant-based proteins, and polysaccharides [16], supporting our finding of higher levels of these bacteria in the individuals who improved their diet drastically. It is widely accepted that the MD is a healthy diet that exerts a beneficial impact on the gut microbiota [37]. However, it was reported that the dietary intake of polyphenols, present in the EVOO, increases the number of bacterial populations of the genera *Bifidobacterium* and *Lactobacillus*, which have anti-pathogenic and anti-inflammatory effects, supporting our findings [27,38]. In contrast, some authors described that polyphenols promote the growth of the species *Akkermansia muciniphila* [39]. In this study, we cannot obtain the bacterial information at species level, but we did not observe any effect of the EVOO and walnut supplementation on the *Akkermansia* genus.

We observed that subjects with a ΔMEDAS score of more than 3 points significantly switched the relative abundance of the *Bifidobacterium* genus (Figure 2b). *Bifidobacterium* has been associated in the literature with the metabolisation of complex carbohydrates, such as vegetal fibre, and is considered one of the main short chain fatty acid (SCFA) producing bacterial genera in the gut [40,41]. This was a solid evidence that a high adherence to the SMD could modify the gut microbiota composition with only twelve weeks of diet intervention. Additionally, we found that the abundance of the *Bifidobacterium* genus was inversely associated with the intake of saturated fats, as described by others, *Bifidobacterium* were correlated with the intake of unsaturated fats [42,43]. This supports our findings and corroborates that a high adherence to the supplementation could modify the gut microbiota.

After confirming that the SMD influenced the composition of the gut microbiota in HIV-1-positive individuals, we explored whether the changes in this composition were related to any changes in the metabolic, inflammatory, microbial translocation, immune activation, and immune tolerance (Treg cells) parameters. We found that specific bacterial genera were influenced by some components of the diet (i.e., calorie intake and carbohydrates) and were inversely associated with *Butyrivibrio* abundances. In a rat model, an increased abundance of the *Butyrivibrio* genus has been associated with diets supplemented with alpha-linoleic acid or prebiotics [44,45], which is consistent with our findings. Our data suggested that some bacterial genera were associated with increased levels of total cholesterol, microbial translocation, and inflammation (*Blautia* and *Catenibacterium*), while others were related to a beneficial profile (the reduction in *Succinivibrio* genus correlated with high immune activation). Interestingly, an increase in *Succinivibrio* abundance has been described in the guts of HIV ART-treated elite controllers compared with ART-naïve HIV-infected individuals [46].

Several studies have reported that the gut microbiota production of SCFA can reduce several inflammatory and allergic diseases [47]. A higher consumption of fruits, vegetables, and legumes is linked to an increase in faecal SCFA levels as well as to higher concentration of microorganisms capable of degrading fibre [48]. Thus, the gut microbiota composition linked with higher MD adherence could be mediating the anti-inflammatory profile observed in these individuals. It is characterised by low immune activation and high levels of Treg cells. They were enriched in fibre-fermenting bacteria, which produce SCFAs, such as butyrate, acetate, and propionate. It was described in the bibliography that these SCFAs induce the stability and integrity of the gut barrier and promote the T-cell differentiation into Tregs [49,50]. This supports our findings, where *Succinivibrio* was associated with the increase of Treg levels. In contrast, we did not observe the expected relationship with the genus *Butyrivibrio* (butyrate-producing bacteria), where we see an inverse association not only with immune activation but also with the Tregs.

This study had a number of potential limitations. First, the evaluated cohort was under a dietary intervention and only the SMD group received nutritional advice. Second, the intervention lasted 12 weeks, and it is likely that longer interventions could have had a different impact on the immunological markers. The quantity of the EVOO and walnuts administered was less than half that used in the previous PREDIMED studies, which could have influenced the results too [14,15]. Furthermore, women were underrepresented in our cohort; therefore, the intervention could have had different effects on the gut microbiota of women.

In conclusion, the Mediterranean diet improved metabolic parameters, immune activation, Treg function, and the gut microbiota composition in HIV-1-infected individuals. Further, the Mediterranean diet increased the *Bifidobacterium* abundances after the intervention, and it was associated to a beneficial profile.

## Figures and Tables

**Figure 1 nutrients-13-01141-f001:**
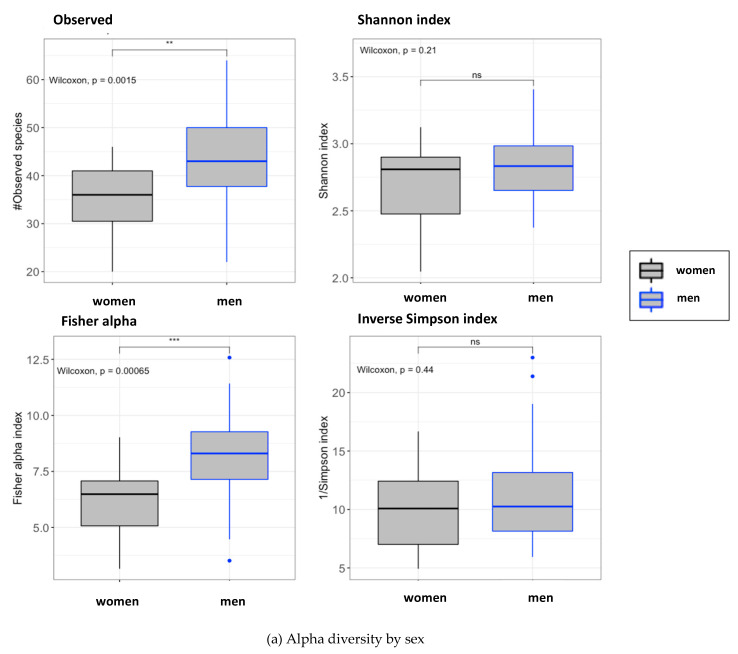
Microbiome diversity and richness in accordance with the sex. (**a**) Alpha diversity according to the sex. Grey boxes correspond to women individuals and blue boxes correspond to men individuals. All subjects from the cohort were selected at the basal time point. Left top: number of observed operational taxonomic units (OTUs); right top: Shannon index; left bottom: Fisher alpha index; right bottom: 1/Simpson index. For every parameter, the p-value for every comparison between women and men is presented. Signification code: *p* < 0.005: ***, *p* < 0.01: **, *p* < 0.05: *, *p* > 0.05: ns. (**b**) Principal coordinate analysis (PCoA) plot clustered by the sex. All individuals included in the cohort were selected and classified by the sex at the basal time point. Non-metric multidimensional scaling (NMDS) used: Bray-Curtis distances. These distances were calculated using the abundances of the bacterial genus’ composition at the basal time point. R package: vegan.

**Figure 2 nutrients-13-01141-f002:**
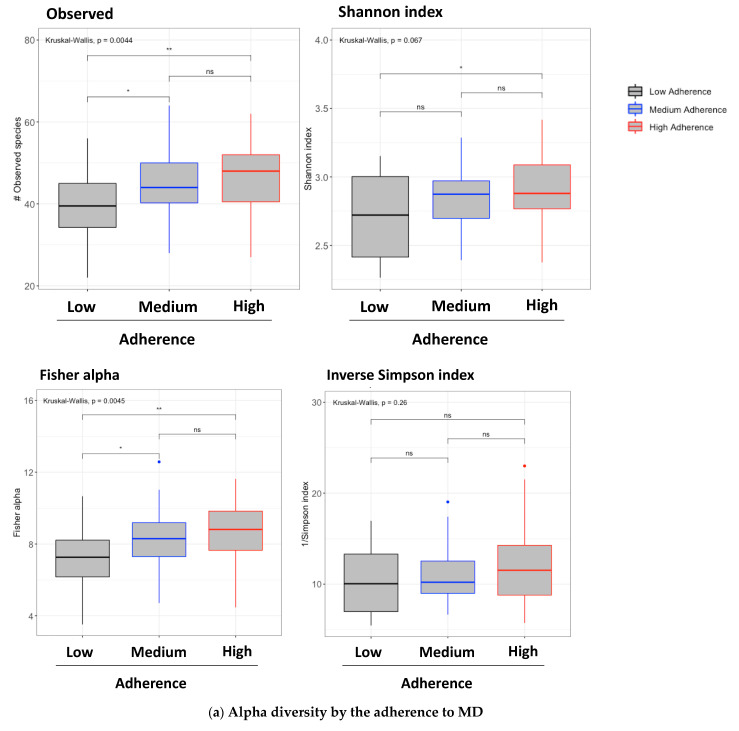
Microbiome diversity and richness in the different adherence groups. (**a**) After classifying the subjects according to their MEDAS scores, alpha diversity was determined at the end of the study. MSM subjects were selected (*n* = 60). Top left: number of observed OTUs; top right: the Shannon index; bottom left: Fisher’s alpha index; and bottom right: Simpson’s diversity index. For each parameter, the global Kruskal–Wallis *p*-value and the *p*-values for every comparison between the adherence groups at week 12 are presented. *** *p* < 0.005, ** *p* < 0.01, * *p* ≤ 0.05. ns: *p* < 0.05. (**b**) PCoA plot clustered by the MEDAS scores from week 12. MSM individuals were selected. Non-metric multidimensional scaling (NMDS) used: Bray–Curtis distances. R package: vegan.

**Figure 3 nutrients-13-01141-f003:**
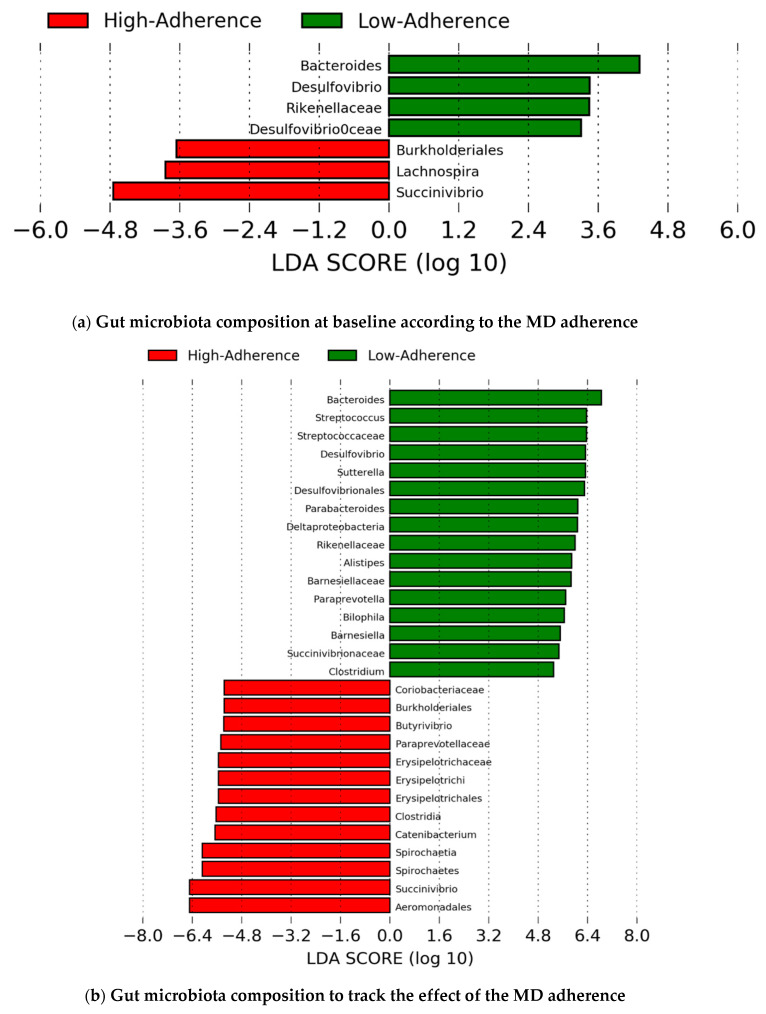
Linear discriminant analysis (LDA) effect size (LEfSe) in the group of MSM individuals. (**a**) Influence of the adherence groups, at baseline (the high-adherence group (*n* = 9) is shown in red, while the low-adherence group (*n* = 19) is shown in green), on the gut microbiota composition. (**b**) Influence of the adherence groups, fixed at week 12 (the high-adherence group (*n* = 30) is shown in red, while the low-adherence group (*n* = 8) is shown in green), on the gut microbiota composition. (**c**) Influence of the increase in the MEDAS score between baseline and week 12 (ΔMEDAS) on the gut microbiota composition (ΔMEDAS ≥ 4 points (*n* = 17): green; ΔMEDAS < 2 points (*n* = 33): red). MSM individuals were selected to avoid bias caused by the sex and the sexual behaviour.

**Figure 4 nutrients-13-01141-f004:**
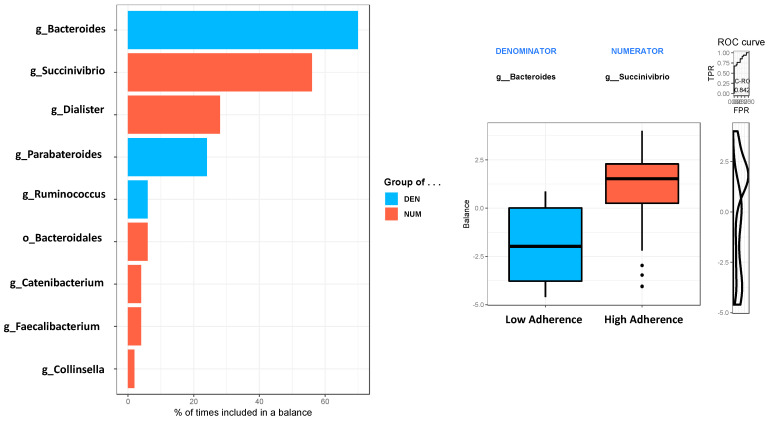
Selbal analysis. This analysis indicated the bacterial genera that best explained the differences between the extreme adherence groups among the MSM individuals (high-adherence group (*n* = 37): red; low-adherence group (*n* = 33): blue). The balance derived from the selbal analysis is shown in the middle of the plot. The cross-validation accuracy of the microbiota classifier is depicted by the receiver operating characteristic (ROC) curve for the bacterial genera obtained, with the area under the curve (top right: AUC = 0.842) indicated inside each plot. As shown in this figure, there was a strong association between the specific microbiome genus composition and the MEDAS classification.

**Figure 5 nutrients-13-01141-f005:**
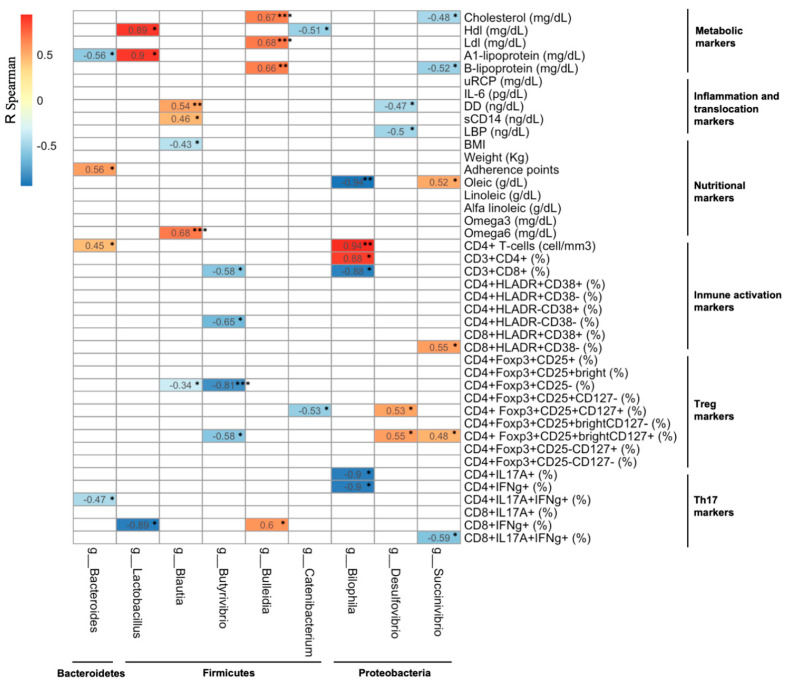
Spearman’s rank correlation coefficients in the high-adherence group at the end of the study. Each column of the table represents a particular bacterial genus, while the rest of the parameters analysed in this study are shown in the rows for the high-adherence group among the MSM individuals at week 12. The bacterial genera (columns) were grouped by phylum (*Bacteroidetes*, *Firmicutes*, and *Proteobacteria*). Only the most significant correlations are shown. Direct (positive) correlations are highlighted in red, while inverse (negative) correlations are highlighted in blue. Test applied: Spearman’s rank correlation with Holm’s correction. The corrected *p*-values are not shown. * *p* < 0.05, ** *p* < 0.01, *** *p* < 0.005. White squares: no significant correlations, *p* ≥ 0.05. The following parameters were analysed: metabolic (cholesterol, HDL, LDL, A1-lipoprotein, and B-lipoprotein), inflammation (uCRP, IL-6, and D-dimer), bacterial translocation (sCD14 and LBP), nutrition (BMI, weight, adherence scores (MEDAS), oleic acid, linoleic acid, alpha-linoleic acid, omega-3, and omega-6), immune activation (CD4+ T-cells, CD3+CD4+, CD3+CD8+, CD4+HLADR, CD8+HLADR), Treg cells (CD4+Foxp3+CD25, Foxp3+CD25+CD127), IL17 and IFN-γ production (CD4+IL17A+, CD4+IFNγ+, CD4+IL17A+IFNγ+, CD8+IL17A+, CD8+IFNγ+, CD8+IL17A+IFNγ+). Acronyms: HDL, high-density lipoprotein; LDL, low-density lipoprotein; uCRP, ultrasensitive C-reactive protein; IL-6, interleukin-6; DD, D-dimer; sCD14, soluble CD14; LBP, lipopolysaccharide-binding protein; Treg, regulatory T-cell; BMI, body mass index.

**Table 1 nutrients-13-01141-t001:** Scores at baseline and at the 12-week follow-up from the 14-item MEDAS questionnaire validated by the PREDIMED guidelines. The percentages of total individuals in each study group (the SMD or control group) are presented in the brackets. The data presented includes all individuals randomised by study group (*n* = 74); men who have sex with men (MSM) randomised by study group (*n* = 60); and a description of the improvement in the MEDAS scores at week 12. The adherence scores of two individuals from the SMD group were not registered at week 12. These two participants were considered as showing no improvements in their MEDAS scores. SMD, supplemented Mediterranean diet; MSM, men who have sex with men; MEDAS, MD Adherence Screener. * Increase. ** Decrease.

Subject Selection	MEDAS Group	Basal (Week 0)	End of Study (Week 12)
**All Individuals (*n* = 74)**			
Control group	High-Adherence	6 (16%)	6 (16%)
(*n* = 37)	(MEDAS ≥ 10)		
	Medium-Adherence	20 (54%)	20 (54%)
	(7 ≤ MEDAS < 10)		
	Low-Adherence	11 (30%)	11 (30%)
	(MEDAS < 7)		
SMD group	High-Adherence	3 (8%)	31 (89%) *
(*n* = 37)	(MEDAS ≥ 10)		
	Medium-Adherence	17 (46%)	4 (11%) **
	(7 ≤ MEDAS < 10)		
	Low-Adherence	17 (46%)	0 **
	(MEDAS < 7)		
**MSM (*n* = 60)**			
Control group	High-Adherence	6 (19%)	6 (19%)
(*n* = 32)	(MEDAS ≥ 10)		
	Medium-Adherence	18 (56%)	18 (56%)
	(7 ≤ MEDAS < 10)		
	Low-Adherence	8 (25%)	8 (25%)
	(MEDAS < 7)		
SMD group	High-Adherence	3 (11%)	24 (92%) *
(*n* = 28)	(MEDAS ≥10)		
	Medium-Adherence	14 (50%)	2 (8%) **
	(7 ≤ MEDAS < 10)		
	Low-Adherence	11 (39%)	0 **
	(MEDAS < 7)		
	**Delta (Δ) MEDAS group**	**Number of subjects**	
**MSM (*n* = 60)**			
Control group	Best improvement, Δ ≥ 4	0	
(*n* = 32)	Δ = 1–3	1 (3%)	
	No improvement, Δ = 0	31 (97%)	
SMD group	Best improvement, Δ ≥ 4	17 (61%)	
(*n* = 28)	Δ = 1–3	9 (32%)	
	No improvement, Δ = 0	2 (7%)

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
