# Peer review of "Adherence to a Supplemented Mediterranean Diet Drives Changes in the Gut Microbiota of HIV-1-Infected Individuals"

_nutrients, 2021, doi:10.3390/nu13041141_

Round 1
Reviewer 1 Report
To the authors
This manuscript described interesting topics about gut microbiota and Mediterranean diet in HIV-1-infected individuals. The authors found that Mediterranean diet changed their gut microbiota and reduced inflammation phenotypes. They also assessed the association between gut microbiota and the phenotypes. I think that the readers are interested in this topic.I have some comments.
- The authors divided their participants into High-adherent and Low-adherent groups to Mediterranean diet at the end of the study and compared gut microbiota. This difference seemed to affect gut microbiota. I think that they could also divide their participants at the basal according to Table 1. I wonder whether the adherence to Mediterranean diet at the basal also affected gut microbiota and inflammatory phenotypes.
- The authors should discuss their results were caused by EVOO and other supplementations themselves or “Mediterranean diet”.
- I think that there should be several PCoA plots for example, High-adherent and Low-adherent groups at basal. Another is four groups which are High-adherent and Low-adherent groups at basal and the end of the study in one figure. The authors should also perform PERMANOVA for the comparison.
- I am interested in the relationship between the change of MEDAS score and the change of gut microbiota.
- I think that the authors should discuss possible mechanism of the effects of gut microbiota changes or Mediterranean diet on the immunological changes observed in this study. Could short chain fatty acids play some roles?
- Line 342: NCD usually means non-communicable diseases.
- Lines 268, 269: Maybe symbols did not appear in PDF version: delta and above?
Author Response
Reviewer 1: This manuscript described interesting topics about gut microbiota and Mediterranean diet in HIV-1-infected individuals. The authors found that Mediterranean diet changed their gut microbiota and reduced inflammation phenotypes. They also assessed the association between gut microbiota and the phenotypes. I think that the readers are interested in this topic. I have some comments.
Thanks to the reviewer for his kind words and suggestions. We address all the concerns of the referees here. Our revisions reflected all reviewers’ suggestions. The added text to the main manuscript was highlighted in red colour. Detailed responses to reviewers are given below.
- The authors divided their participants into High-adherent and Low-adherent groups to Mediterranean diet at the end of the study and compared gut microbiota. This difference seemed to affect gut microbiota. I think that they could also divide their participants at the basal according to Table 1. I wonder whether the adherence to Mediterranean diet at the basal also affected gut microbiota and inflammatory phenotypes.
We understand the reviewer’s comment about how the adherence to Mediterranean diet (MD) affect the gut microbiota composition at the beginning of the study. At baseline, we observed significant differences in the gut microbiota composition according to the nutritional behaviour expressed as adherence score provided by the validated 14-item MD Adherence Screener (MEDAS) from the PREDIMED study. These differences were accentuated after the dietary intervention, because the number of subjects included in the High-Adherence group significantly increased, as we mentioned in the main text (line 220-226). According to the reviewer suggestion, we included the figure presented below, in the main text as Figure 3a (line 353) and the corresponding explanatory text in the manuscript (lines 240-241).
Figure 3. Linear discriminant analysis (LDA) effect size (LEfSe) in the group of MSM individuals. (a) Influence of the adherence groups, at baseline (the High-Adherence group (n=9) is shown in red, while the Low-Adherence group (n=19) is shown in green), on the gut microbiota composition. MSM individuals were selected to avoid bias caused by the sex and the sexual behaviour.
- Gut microbiota composition at baseline according to the MD adherence
Regarding the inflammatory phenotypes, at baseline, we did not observe different inflammatory profile or T-cell subset activation between groups. Thus, we have added the table (showed below) with the basal comparisons between the MD adherence groups, according with the reviewer consideration, in the Supplementary data (Supplementary Table 4) and the corresponding explanatory text in the manuscript (lines 290-293). In addition, you can find the comparisons for the all-analysed markers between the different time-points (and the delta) for each MD adherence group in the Supplementary Table 5.
Supplementary Table 4. Inflammatory, bacterial translocation and immunological markers in High-Adherence (n=9) and Low-Adherence (n=31) groups at baseline. Mann-Whitney-Wilcoxon test with Bonferroni was used to examine parameters between Adherence extreme groups. P<0.05 was considered significant; not significantly significant features between groups were detected. Acronyms: uRCP, ultrasensitive c-reactive protein; IL-6, interleukin-6; DD, D-dimer; sCD14, soluble CD14; LBP, lipopolysaccharide binding-protein; Treg, regulatory T-cell.
|
Marker |
Low-Adherence at baseline |
High-Adherence at baseline |
P-value |
|
|||
|
Inflammation |
|
|
|
|
|
|
|
|
PCR |
0.27 |
0.22 |
0.473 |
|
|||
|
IL6 |
3.90 |
3.67 |
0.438 |
|
|||
|
DD |
253.33 |
422.22 |
0.107 |
|
|||
|
Bacterial translocation |
|
|
|
|
|||
|
sCD14 |
1735.39 |
1653.89 |
0.987 |
|
|||
|
LBP |
11417.76 |
13081.04 |
0.180 |
|
|||
|
Immune activation |
|
|
|
|
|||
|
CD4+ (%) |
48.28 |
47.99 |
0.897 |
|
|||
|
CD8+ (%) |
43.41 |
44.40 |
0.799 |
|
|||
|
CD4+HLADR+CD38+ (%) |
2.25 |
1.65 |
0.750 |
|
|||
|
CD8+HLADR+CD38+ (%) |
3.65 |
4.12 |
0.799 |
|
|||
|
Treg cells |
|
|
|
|
|||
|
CD4+Foxp3+CD25+ (%) |
4.65 |
3.94 |
0.206 |
|
|||
|
CD4+Foxp3+CD25+bright (%) |
2.14 |
1.86 |
0.507 |
|
|||
|
CD4+Foxp3+CD25- (%) |
3.78 |
3.71 |
0.373 |
|
|||
|
CD4+Foxp3+CD25+CD127- (%) |
4.06 |
3.41 |
0.199 |
|
|||
|
CD4+Foxp3+CD25+CD127+ (%) |
0.59 |
0.48 |
0.248 |
|
|||
|
CD4+Foxp3+CD25+brightCD127- (%) |
1.96 |
1.68 |
0.371 |
|
|||
|
CD4+Foxp3+CD25+brightCD127+ (%) |
0.18 |
0.16 |
0.639 |
|
|||
|
T-helper 17 (Th17) cells |
|
|
|
|
|||
|
CD4+IL17A+ (%) |
0.84 |
0.71 |
0.354 |
|
|||
|
CD4+IFNg+ (%) |
18.40 |
25.86 |
0.093 |
|
|||
|
CD4+IL17A+IFNg+ (%) |
0.15 |
0.17 |
0.891 |
|
|||
|
CD8+IL17A+ (%) |
0.02 |
0.02 |
0.830 |
|
|||
|
CD8+IFNg+ (%) |
58.60 |
57.46 |
0.711 |
|
|||
|
CD8+IL17A+IFNg+ (%) |
0.05 |
0.03 |
0.355 |
|
|||
- The authors should discuss their results were caused by EVOO and other supplementations themselves or “Mediterranean diet”.
We appreciate the reviewer comments about the discussion of EVOO effects on the human health. In this regard, we provided a brief discussion of the manuscript. In order to enlarge this section, we added the following information:
(line 428-439) ‘’In this study, individuals who supplemented their diet with EVOO and walnuts improved their lipid profile, reducing their total cholesterol and B-lipoprotein levels. Healthy benefits associated to MD may be explained by its great content in polyphenols. Polyphenols play an important role in the health promoting, as they seem to show anti-inflammatory, antioxidant, antidiabetic or anticarcinogenic effects, as well as, improve lipid profile or adiposity. In the small intestine the absorption of polyphenols is low (<10%), and the rest of polyphenols can positively influence in the microbiota composition. Nevertheless, which are the underlaying mechanisms through polyphenols can exert these positive effects are unknown yet [17,24,25]. There is to highlighted, that in our study, individuals with a high adherence to the MD, with high EVOO and walnuts intake, improved their lipid profile, reducing their total cholesterol and B-lipoprotein levels. EVOO contains polyphenols which possess vasodilatory, anti-thrombotic, anti-inflammatory and anti-apoptotic effects, leading to a cardioprotective state, as well as anti-lipemic and anti-atherogenic effects [26,27]. ‘’
(line 461-467) ‘’It's widely accepted that MD is a healthy diet that exerts a beneficial impact on the gut microbiota [37]. But, it was reported that the dietary intake of polyphenols, present in the EVOO, increases the number of bacterial populations of the genus Bifidobacterium and Lactobacillus, which have anti-pathogenic and anti-inflammatory effects, supporting our findings [27,38]. In contrast, some authors described that polyphenols promote the growth of specie Akkermansia muciphila [39]. In this study, we cannot obtain the bacterial information at species level, but we did not observe any effect of the EVOO and walnuts supplementation on the Akkermansia genus.’’
- I think that there should be several PCoA plots for example, High-adherent and Low-adherent groups at basal. Another is four groups which are High-adherent and Low-adherent groups at basal and the end of the study in one figure. The authors should also perform PERMANOVA for the comparison.
We really appreciate the reviewer recommendation about the PCoA plots. Here we present the PCoA suggested by the reviewer, both for the entire cohort (a and b plots showed below) and for the sub-study with MSM (c and d plots showed below). We added the these PCoA to the Supplementary data (Supplementary Figure 2) in accordance with the reviewer comments.
Supplementary Figure 2. Principal Coordinates Analysis (PCoA). Non-metric multidimensional scaling (NMDS) used: Bray Curtis distances. R package: vegan. a) Complete cohort adherence at basal time-point. The Low-adherence (LA) group is represented by the black squares, n=31; the High-Adherence (HA) group is represented by the red squares, n=9. The Adonis (PERMANOVA) test was performed, considering the adherence extreme groups (High and Low adherence): P=0.145, R2=0.045. b) Complete cohort by MD adherence at baseline and at the end of the study. The Low-adherence group at baseline (LA_basal) is represented by the black squares, n=31; the Low-adherence at the end of the study (LA_end) group is represented by the blue squares, n=11; the High-Adherence group (HA_basal) is represented by the red squares, n=9; The High-adherence at the end of the study (HA_end) group is represented by the orange squares, n=37. The Adonis (PERMANOVA) test was performed, considering the MD adherence groups (High and Low adherence): P=0.061, R2=0.060. c) MSM individuals by the adherence behaviour to MD at baseline time-point. The Low-adherence (LA) group is represented by the black squares, n=19; the High-Adherence (HA) group is represented by the red squares, n=9. The Adonis (PERMANOVA) test was performed, considering the adherence extreme groups (High and Low adherence): P=0.108, R2=0.076. d) MSM individuals by the adherence behaviour to MD at baseline and at the end of the study. The Low-adherence group at baseline (LA_basal) is represented by the black squares, n=19; the Low-adherence at the end of the study (LA_end) group is represented by the blue squares, n=8; the High-Adherence group (HA_basal) is represented by the red squares, n=9; The High-adherence at the end of the study (HA_end) group is represented by the orange squares, n=30. The Adonis (PERMANOVA) test was performed, considering the adherence extreme groups (High and Low adherence): P=0.197, R2=0.074.
Due to the different number of patients included in the High and Low Adherence group at baseline and at the end of the study, and in order to avoid bias induced by the inclusion of the same subject in two different groups, we stablished the adherence groups by the MEDAS score at the end of the study (week 12), thus achieving greater statistical power.
- I am interested in the relationship between the change of MEDAS score and the change of gut microbiota.
We welcome the reviewer comment on the relationship between the change of MEDAS score and the change of gut microbiota. We added a brief discussion of the mentioned relationship in the discussion section.
In order to track the direct changes induced by the supplemented Mediterranean diet and to avoid the effect of subjects with a good basal nutritional behaviour, we compared those who increased their MEDAS score more than 3 points during the study (all from the SMD group) and subjects who did not change in their MEDAS score during the study (all from control group).
(lines 468-472) ‘’We observed that subjects with a DMEDAS score more than 3 points switch significatively the relative abundance of Bifidobacterium genus (Figure 2b). Bifidobacterium has been associated in the literature with the metabolization of complex carbohydrates, as vegetal fibre, and is considered one of the main SCFA-producing bacterial genera in the gut [40,41]. This was a solid evidence that a high adherence to the SMD could modify the gut microbiota composition with only twelve weeks of diet intervention.’’
We also performed the Spearman rank correlation also with those subjects who increased their MEDAS score more than 3 points during the study. We tried to associate the switch in the gut microbiota populations with changes in the analysed markers. (lines 472-476) “Additionally, we found that the abundance of the Bifidobacterium genus was inversely associated with the intake of saturated fats, as described others, Bifidobacterium were correlated with the intake of unsaturated fats [42,43]. It supports our findings and corroborate that a high adherence to the supplementation could modify the gut microbiota.’’
- I think that the authors should discuss possible mechanism of the effects of gut microbiota changes or Mediterranean diet on the immunological changes observed in this study. Could short chain fatty acids play some roles?
We agree with the reviewer's warning about the lack of discussion of the possible mechanism of the effects of changes in gut microbiota on the immunological changes observed in this study, and we really appreciate the suggestion of discuss the role of the SCFA. To solve this issue, we added the following lines to the manuscript discussion section.
(lines 489-499) ‘’Several studies have reported that the gut microbiota production of SCFA can reduce several inflammatory and allergic disease [47]. A higher consumption of fruit, vegetables and legumes is linked to an increase in faecal SCFA levels as well as to higher concentration of microorganisms capable of degrading fibre [48]. Thus, the gut microbiota composition linked with higher MD adherence could be mediating the anti-inflammatory profile observed in these individuals. It is characterised by low immune activation and high levels of Treg cells. They were enriched in fibre-fermenting bacteria which considers SCFA-producers, as butyrate, acetate and propionate. It was described in the bibliography that this SCFA induce the stability and integrity of the gut barrier and promote the T-cell differentiation into Treg [49,50]. This supports our findings, where Succinivibrio was associated with the increase of Treg levels. In contrast, we do not observe the expected relationship with the genus Butyrivibrio(butyrate-producing bacteria) where we see an inversely association with immune activation, but also with the Treg.’’
- Line 342: NCD usually means non-communicable diseases.
Thank you for the clarification, we have corrected it.
- Lines 268, 269: Maybe symbols did not appear in PDF version: delta and above?
Thank you very much for noted this typo error, we have corrected it.

Reviewer 2 Report
In the manuscript titled “Adherence to a supplemented Mediterranean diet drives changes in the gut microbiota of HIV-1-infected individuals”, Authors aimed to test hypothesis that the benefits of a supplemented Mediterranean diet (SMD) could be mediated by changes in the gut microbiota, even in those with an altered intestinal microbiota such as people living with HIV. This is an important topic because microbiome plays an important role in metabolism and immune system. The following comments need to be addressed in the presented study:
Methods:
The methods are presented in very unspecific way. It is hard to follow which test was done at what point and how many subjects were tested. The information about the baseline and end point status of all subjects is lacking critical data (sex, body weight, body composition, age, level of circulating cytokines and microbiome profile). Schematic representation of experimental design would significantly improve the understanding of the study.
Results:
The figures presenting the studied factors at the baseline compared to the end point of the experiment within the control and SMD group as well as between the groups at each point are missing. The result section should present only results obtained from subjects participating in all stages of the experiment. The figure/s presenting sex differences should be presented in the main manuscript.
Discussion:
The major findings of the study should be highlighted and properly discussed. The way results are presented is not sufficient to do support the conclusions.
Minor:
The language should be revised. Gender is a social concept and use of sex (male subjects; female subjects) is proper in biomedical research.
Author Response
Reviewer 2: In the manuscript titled “Adherence to a supplemented Mediterranean diet drives changes in the gut microbiota of HIV-1-infected individuals”, Authors aimed to test hypothesis that the benefits of a supplemented Mediterranean diet (SMD) could be mediated by changes in the gut microbiota, even in those with an altered intestinal microbiota such as people living with HIV. This is an important topic because microbiome plays an important role in metabolism and immune system. The following comments need to be addressed in the presented study:
Thanks to the reviewer for his comments and suggestions. We address all the concerns of the referees here. Our revisions reflected all reviewers’ suggestions. The added text to the main manuscript was highlighted in red colour. Detailed responses to reviewers are given below.
Methods: 1) The methods are presented in very unspecific way. It is hard to follow which test was done at what point and how many subjects were tested. 2) The information about the baseline and end point status of all subjects is lacking critical data (sex, body weight, body composition, age, level of circulating cytokines and microbiome profile). 3) Schematic representation of experimental design would significantly improve the understanding of the study.
- The methods are presented in very unspecific way. It is hard to follow which test was done at what point and how many subjects were tested
We acknowledged the reviewer’s suggestion. To improve Methods clarity, the bioinformatics and statistical analysis section (starting at line 157) has been rewritten, as shown in the reviewed manuscript version and every issue noticed have been addressed, hoping it is properly specified now.
- The information about the baseline and end point status of all subjects is lacking critical data (sex, body weight, body composition, age, level of circulating cytokines and microbiome profile)
We agree with the reviewer’s comments. To provide the information data requested and additional details, we include Supplementary documents:
- Supplementary Table 1: Basal values of sex, body weight as body mass index (BMI), BMI classification, age, origin, risk group (MSM or no-MSM), antiretroviral drugs treatment (ART), years on ART, CD4 T-cell count and, nadir are provided. No statistical differences were found for any of the shown parameters between randomized groups.
- Supplementary Table 2. Metabolic, inflammation, bacterial translocation, immunological and nutrition markers in Supplemented Mediterranean Diet group (SMD) and control group are provided. It was compared both time-points resulting from each randomized group (basal vs week 12) and Δ from each randomized group (ΔSMD vs Δcontrols). The P-values at basal between randomization groups were not showed due to lack of statistical significance.
- Supplementary Table 4 (added after revision) provides basal comparisons between the MD adherence groups as suggested.
- Supplementary Table 5 contains comparisons for total-analysed markers between the different time-points considered (as well as Δ value) for each MD adherence group.
If you consider that some of this tables should be included in the main text, we can incorporate them.
3) Schematic representation of experimental design would significantly improve the understanding of the study.
As the reviewers suggest, we included the schematic description of participants follow-up in the Supplementary Figure 1. We also provide a schematic graphical abstract (GA) to improve the understanding of the study. The cited Supplementary Figure 1 and the GA are presented below.
Supplementary Figure 1. Schematic description of participants follow-up.
Graphical Abstract (GA):
Results: 1) The figures presenting the studied factors at the baseline compared to the end point of the experiment within the control and SMD group as well as between the groups at each point are missing. 2) The result section should present only results obtained from subjects participating in all stages of the experiment. 3) The figure/s presenting sex differences should be presented in the main manuscript.
1) The figures presenting the studied factors at the baseline compared to the end point of the experiment within the control and SMD group as well as between the groups at each point are missing.
We appreciate this suggestion. In fact, the studied factors (metabolic, microbial translocation, inflammatory, nutritional, and immunological markers) were analysed at different time-points both for the randomization groups and the MD adherence. You can find the complete results in the Supplementary Tables 2 to 5. If you consider that this information should be provided by a figure, we could select the statistically significant data and show it as a box plot or something similar. Or in the other hand, if you consider that some of this tables should be included in the main text, we can incorporate them.
2) The result section should present only results obtained from subjects participating in all stages of the experiment.
We agree with the reviewer comment. We presented the obtained results in the first part of the study, when analysed the Randomization groups. For this purpose, all individuals who reach the end of the study were analysed. We think that the Supplementary Figure 1 could be useful to track the patients used for each comparison.
3) The figure/s presenting sex differences should be presented in the main manuscript
As the reviewer suggests, we have incorporated the figures presenting sex differences in the main text of the manuscript (lines 256-267).
Discussion: 1) The major findings of the study should be highlighted and properly discussed. 2) The way results are presented is not sufficient to do support the conclusions.
1) The major findings of the study should be highlighted and properly discussed
We are grateful to receive your comments about the discussion of this work. In order to clarify the main findings of the study we numerate them in the following list:
- The SMD group improved their lipidic profile, the immune activation and IFN-ɣ-producing T-cells in the SMD group after twelve weeks of intervention (Supplementary Table 2).
- A high adherence to a MD increases the alpha diversity (Figure2) and the prevalence of Burkholderiales, Butyrivibrio, Catenibacterium, and Succinivibrio in the gut microbiota composition (Figure 3b and Figure 4). This bacterial profile is associated with a reduction in the immune activation (Figure 5).
- The switch in the nutritional behaviour (measured by the ΔMEDAS) from a Western diet to Mediterranean diet increases the Bifidobacterium abundances in the gut microbiota (Figure 3c). This change was associated with changes in the Treg cells (Supplementary Figure 5).
According to the reviewer's concern about how the results have being discussed, we also extended the discussion section to properly discuss the major findings of the study (it was highlighted in red colour in the main text).
2) The way results are presented is not sufficient to do support the conclusions.
We value the reviewers' comments very much. We have modified the conclusions so as to fit better with the presented results:
(lines 507-509) ‘’In conclusion, the Mediterranean diet improved metabolic parameters, immune activation, Treg function and the gut microbiota composition in HIV-1-infected individuals. Further, Mediterranean diet increased the Bifidobacterium abundances after the intervention and it was associated to a beneficial profile.’’
Minor: The language should be revised. Gender is a social concept and use of sex (male subjects; female subjects) is proper in biomedical research.
A native English speaker has checked the whole manuscript grammar trying to solve the language problem observed by the reviewer.
As pointed out we have replaced gender by sex through the manuscript.